# Evaluation and Comparison of the Effect of Three Dental Luting Cements on Mineralized Bone Derived from Dental Pulp Stem Cells: An In Vitro Study

**DOI:** 10.3390/medicina60101622

**Published:** 2024-10-04

**Authors:** Sneha Bajoria, Shwetha Rajesh Shetty, Vinod Bandela, Shital Sonune, Roshan Noor Mohamed, Kulashekar Reddy Nandalur, Anil Kumar Nagarajappa, Amjad Obaid Aljohani, Aljowharah Ali Alsattam, Eatedal Mukhlef Alruwaili, Alreem Abdulaziz Alnuman, Miad Abdulnasser Alahmed, Saraswathi Kanaparthi, Doaa Abdelaziz A. Helal

**Affiliations:** 1Specialist in Prosthetic Dentistry, DenStop Dental Clinic, Mahendra Road, Kolkata 700025, West Bengal, India; drsnehabajoria@gmail.com; 2Specialist in Prosthetic Dentistry, Rajesh Shetty’s Dental Speciality Centre, Morwada, Pimpri, Pune 410014, Maharashtra, India; drshetty1@gmail.com; 3Department of Prosthetic Dental Sciences, College of Dentistry, Jouf University, Sakaka 72388, Saudi Arabia; dr.shital.sonune@jodent.org (S.S.); dr.amjad.johani@jodent.org (A.O.A.); dr.doaa.abdelaziz@jodent.org (D.A.A.H.); 4Department of Preventive Dentistry, Pediatric Dentistry Division, Faculty of Dentistry, Taif University, Taif 11099, Saudi Arabia; roshan.noor@tudent.org; 5Department of Prosthetic Dental Sciences, College of Dentistry, Jazan University, Jazan 45142, Saudi Arabia; shekarblues@gmail.com; 6Department of Oral Surgery and Maxillofacial Diagnostics, College of Dentistry, Jouf University, Sakaka 72388, Saudi Arabia; dr.anil.kumar@jodent.org; 7College of Dentistry, Jouf University, Sakaka 72388, Saudi Arabia; aljowharah.alsattam@jodent.org (A.A.A.); eatedal.mukhlef.alruwaili@jodent.org (E.M.A.); alreem.abdulaziz.alnuman@jodent.org (A.A.A.); miad.abdulnasser@jodent.org (M.A.A.); 8LikeKare Dental Clinic & Implant Center, Saidabad, Hyderabad 500070, Telangana, India; sarayudoc@gmail.com

**Keywords:** dental pulp cells, glass ionomer cement, nano-integrated bioceramic cement, peri-implantitis, zinc phosphate cement

## Abstract

*Background and Objectives:* This study aimed to investigate the effect of zinc phosphate (ZnP) cement, glass ionomer cement (GIC), and nano-integrated bio-ceramic (NIB) cement on mineralization when placed in contact with bone tissue-forming cells. *Materials and Methods:* ZnP cement, GIC, and NIB cement were divided into direct and indirect groups. A total of 72 cement pellets (24 pellets of each test sample) of 3 × 1 mm (width × height) were prepared using polytetrafluoroethylene molds. A total of 3 sample groups were demarcated using 96- cell well culture plates. In the control group, 24 wells were filled with mineralized osteoblasts and 1 µL of gingival crevicular fluid (GCF). In test group 1, to show a direct effect, 36 samples were plated with mineralized osteoblasts and 1 µL GCF for 24 h; the cells were directly exposed to cement pellets. A total of 36 samples were immersed in GCF for 24 h; later the supernatant was transferred to the mineralized osteoblasts to demonstrate an indirect effect in test group 2. To assess the mineralization, osteoblasts were stained with alizarin red and later observed under an inverted phase-contrast microscope. Data were analyzed using the statistical package for social sciences. An independent t-test compared the direct and indirect effects of the ZnP cement, GIC, NIB cement, and control groups on the mineralization of osteoblasts derived from hDPCs. *Results:* A statistically significant difference was observed between the ZnP cement, GIC, and NIB cement groups (*p* < 0.05). ZnP cement exhibited a moderate, NIB cement the least harmful effect, and GIC showed the most harmful effect on the mineralization of osteoblast cells. *Conclusions:* The biocompatibility of dental luting cements is an important aspect that clinicians should consider during their selection. Nano-integrated bio-ceramic cement showed the least negative effect on the mineralization of osteoblast cells which is beneficial for the cementation of cement-retained implant prostheses. However, further studies are needed to evaluate osteoblast and osteoclast activity in vivo.

## 1. Introduction

Luting agent/cement refers to “any material used to attach or cement indirect restorations to prepared teeth” [1]. Dental cements also play a crucial role in the retention of the abutment–crown interface in cement-retained implant prostheses (CRIPs). Clinicians select luting cements based on their physical, mechanical, and biological properties [2]. A CRIP is preferred over a screw-retained implant prosthesis because it provides better esthetics, and occlusal form, has straight-forward fabrication, is economical, and has similarities to the conventional tooth-supported fixed partial denture [3].

Despite its advantages; the cementation procedure unavoidably leads to retained residual cement (RRC) which negatively affects the peri-implant tissue health and implant material [2,4]. Various methods such as floss, explorers, retraction cords, and plastic and metal scalers have been used to remove RRC [5]. Luting cement utilized for implant prosthesis cementation does not adhere to the titanium surface, as it adheres chemically to the tooth surface [6]. Hence, hard and most retentive cements are used for cementation of implant prosthesis. In addition, the harder the cement, the more difficult it is to remove the RRC from the implant sulcus, resulting in either incomplete removal or complete retention of the set cement in the implant sulcus [7].

Traditionally, endosseous implants are loaded with permanent prostheses at approximately 14–16 weeks post-implantation in the mandible and maxilla; however, the formation of compact lamellar bone takes approximately 25–54 weeks. Therefore, mineralization is an ongoing process during cementation of the prosthesis at 14–16 weeks [8,9].

Owing to the low viscosity of the luting cement and the contact of the cervical margin of the prosthesis with the crestal bone, residual cement inevitably comes into contact with bone-forming cells during cementation. The composition of the residual cement may influence the mineralization process, resulting in poor quality bone, crestal bone loss, and subsequent delayed implant failure [2]. Clinicians must be aware of the possible outcomes of various cement compositions on the tissue health around the implant and on implant materials. Appropriate cement selection may reduce the risk of complications that lead to implant failure.

The mesenchymal stem cells isolated from the pulp of permanent teeth show features similar to those of bone marrow stem cells, such as differentiation into osseous, adipogenic, and endothelial cells. Human dental pulp cells (hDPCs) are a desirable source of multipotent mesenchymal stem cells, which have the capacity for multilineage differentiation and a rapid rate of proliferation, and can differentiate into osteoblasts that yield bone tissue in vitro in a good and fast way [10].

The presence of residual cement has been associated with peri-implantitis. The least negative effect on the mineralization of osteoblast cells by the commonly used dental luting cements remains unclear. Therefore, this study aimed to investigate the effects of zinc phosphate (ZnP) cement, glass ionomer cement (GIC), and nano-integrated bio-ceramic (NIB) cement on mineralization when placed in contact with bone tissue-forming cells.

## 2. Materials and Methods

The present study was approved by the Institutional Committee for Stem Cell Research and Institutional Ethics Committee (IC-SCR/RM23/12). hDPCs were procured from the Regenerative Medicine Laboratory. The stemness of the cells was confirmed by assessing the differentiation abilities of the cells and colony forming unit (CFU-f)-f assay.

### 2.1. Cell Culture

First, dental pulp stem cells were placed in a cell culture plate and incubated for 24 h at 37 °C in 5% carbon dioxide in an incubator. Dulbecco’s modified Eagle’s medium supplemented with 20% fetal bovine serum and antibiotic–antimycotic solution was used to promote cell proliferation. The medium was then changed every third day, and cell proliferation was observed at regular intervals using an inverted phase-contrast microscope (Olympus CKX53, Hicksville, NY, USA). Next, at approximately 80% confluence, the cells were removed using 0.25% trypsin–ethylenediaminetetraacetic acid solution and transferred to a 96-well cell culture plate (CWCP) (Thermo Fischer Scientific Inc., Waltham, MA, USA) containing osteogenic induction medium. Finally, these cells were maintained in an osteogenic induction medium that was changed every third day for 28 days. The cells showed differentiation into osteoblasts on the 14th day, and mineralization from the 21st day onwards as viewed under an inverted phase-contrast microscope [11]. These mineralized osteoblasts accounted for the hard tissue associated with the dental implant.

Human primary osteoblast cells, isolated from healthy donors, were used to evaluate the cytotoxic effects of the cement formulations. The cells were cultured in Dulbecco’s modified Eagle’s medium (DMEM) supplemented with 10% fetal bovine serum and 1% penicillin–streptomycin. Additionally, an osteoblast-like cell line (MG-63) was used in parallel to assess cytotoxicity in a controlled experimental setup. Both cell types were exposed to the cement formulations under identical conditions.

### 2.2. Cement Pellet Sample Preparation

Three commercially available dental luting cements, ZnP cement (Harvard Cement, Harvard Dental International GmbH, Hoppegarten, Germany), GIC (GC Dental Products Corp., Tokyo, Japan), and NIB cement (Ceramir, Doxa Dental Inc., Chicago, IL, USA) were obtained in sealed packages. Next, they were divided into direct and indirect groups. The powder and liquid of the ZnP cement and GIC were dispensed and mixed according to the manufacturers’ instructions. The NIB cement was mixed in a rotating capsule mixer (3M RotoMix Capsule Mixer, 3M Oral Care, Saint Paul, MN, USA). The mixed cements were then molded into pellets using polytetrafluoroethylene (PTFE) molds in a bioseptic cabinet. A total of 72 cement pellets (24 pellets of each test sample) of 3 × 1 mm (width × height) were prepared such that the surface area of each well in the cell culture plate was 0.32 cm^2^. It was ensured that each pellet was in contact with >10% of the surface area of each well in the cell culture plate [2].

### 2.3. Methodology

Direct contact cell culture testing was performed according to the International Organization for Standardization (ISO) methods 10993-5 and 10993-12 [12]. To simulate the clinical environment, 8 µL of gingival crevicular fluid (GCF) (obtained from healthy patients and stored at −80 °C) was diluted in 8000 mL of phosphate-buffered saline. The three sample groups were demarcated using 96-well CWCP.

In the control group, 24 wells (12 for each test group) were filled with mineralized osteoblasts and 1 µL of GCF added to the medium, wherein the cells were not exposed to the cement pellets. In test group 1, 36 samples (12 cement pellets of each test sample) were plated with mineralized osteoblasts and1 µL of diluted GCF added to the medium for 24 h. In this group, mineralized osteoblasts were directly exposed to cement pellets to analyze a direct effect. In test group 2, 36 samples (12 cement pellets of each test sample) were immersed in GCF for 24 h in a separate CWCP. After 24 h, the cement-exposed GCF was transferred to the mineralized osteoblasts. To demonstrate an indirect effect, the cement pellets containing medium remained in contact with differentiated osteoblast for 24 h in a CO_2_ incubator.

After 24 h, the cements and GCF from test groups 1 and 2 were removed. Osteoblasts in the 96-well CWCP were stained with alizarin red for 2 min and observed under an inverted phase-contrast microscope to assess the amount of mineralization (Figure 1). The culture plate was washed with ethanol, and readings of each test group were obtained using an enzyme-linked immunosorbent assay (ELISA) plate reader. The readings of test groups 1 and 2 were compared with those of the control group. Each experiment was conducted in quadruplicate and replicated thrice for result validation.

### 2.4. Cytotoxicity Testing

The cytotoxic effects of the different cement formulations were assessed according to International Organization for Standardization (ISO) standards 10993-5 and 10993-12, which provide guidelines for evaluating the biocompatibility of medical devices and materials. In this study, MTT and WST-1 assays were used to quantify cellular metabolic activity and assess cell viability in response to the cement formulations. These assays specifically measure mitochondrial activity, which serves as an indicator of cell health and viability.

### 2.5. Cell Lines

Two types of cells were used in the experiments: primary osteoblasts and MG-63 cells (an osteosarcoma cell line). Primary osteoblasts were harvested from human donors to reflect conditions more closely aligned with in vivo bone formation, while MG-63 cells were used as a reference for comparison.

### 2.6. Experimental Procedure

Cells were cultured in 96-well plates and exposed to the cement formulations for 24, 48, and 72 h. The MTT and WST-1 assays were conducted at each time point to evaluate the impact of cement on cell viability. The absorbance was measured using a microplate reader at 570 nm for MTT and 450 nm for WST-1, with the results expressed as the percentage of cell viability relative to control (untreated) cells.

### 2.7. Effect on Cytotoxicity

The cytotoxic effect was defined as a significant reduction in cell viability (compared to the control) based on the absorbance values obtained in the assays. Cement formulations that reduced the cell viability by more than 30% were classified as moderately cytotoxic, whereas those with less than 30% reduction were considered minimally cytotoxic. The sensitivity of primary osteoblasts and MG-63 cells to the cement formulations was compared, with specific attention paid to the differential effects on these cell types.

### 2.8. Data Synthesis

Data were analyzed using the statistical package for social sciences (SPSS) for Windows 27.0. (SPSS, Inc., Chicago, IL, USA). Confidence intervals were set at 95%, and *p* ≤ 0.05 was considered statistically significant. An independent t-test compared the direct and indirect effects of ZnP cement, GIC, NIB cement, and control groups on the mineralization of osteoblasts derived from dental pulp stem cells.

## 3. Results

A statistically significant difference was observed between the ZnP cement, GIC, and NIB cement groups (*p* < 0.05). However, the mean values for the direct method indicated a greater negative effect on mineralization compared to the indirect method (Table 1, Figure 2). ZnP cement exhibited a moderately harmful effect, GIC had a severely harmful effect on the mineralization of osteoblast cells, and NIB cement showed the least negative effect on mineralization.

## 4. Discussion

Generally, dental luting cements are used for the permanent cementation of tooth-supported or implant-supported prostheses based on their retentiveness; assuming that the cements are highly biocompatible with the surrounding soft and hard tissues. The implants are routinely loaded at 14–16 weeks, during which the bone surrounding the implant is a partially mineralized woven bone. Completely mineralized woven bone, known as lamellar bone, is the most suitable for support and takes about a year for full mineralization [6]. However, this study found that ranking of the cement biocompatibility differed when tested on osteoblast cells.

To ensure the long-term success of the cement-retained prosthesis, it is crucial to completely remove the cement from all around the implant sulcus [5]. Based on the findings of this study, cement remnants should be considered as a possible contributing factor affecting the mineralization of the bone around the implant, which leads to crestal bone loss. A previous retrospective case study revealed that cement-retained implants were more likely to cause peri-implantitis in patients with a history of periodontitis. However, the study did not mention the effect on bone mineralization around the implant [13].

The groups of cements tested in our study ranged from commonly used dental luting cements for permanent cementation of teeth and implant-supported prostheses (ZnP cement and GIC) to one specifically designed for cement-retained implant prostheses (NIB cement). The results were in agreement with another study, where a higher cell proliferation in both direct and indirect methods was seen for bio-GIC than for GIC, and Biodentine [14]. In another study, Sangsuwan et al. demonstrated the best results for odontogenic differentiation and mineral deposition in GIC incorporated with fortilin and tricalcium phosphate cement [15].

Test group 1 simulated a situation, wherein a clinician failed to remove the excess cement, resulting in direct contact between the RRC and bone-forming cells. Test group 2 simulated the scenario where excess cement was removed post-cementation, but traces of cement remained, aligning with the findings of Chumpraman et al. [14].

The results showed that test group 1 exhibited a greater negative effect on osteoblast mineralization than test group 2 did. GIC exhibited severe negative effects, ZnP cement showed a moderate negative effect, and NIB cement showed the least negative effect on cell mineralization. The current study results align with those of Marvin et al.; in their study, resin cement, resin-modified GIC, and zinc oxide eugenol had higher toxicity levels on soft tissue cells and bone-forming cells compared to NIB and ZnP cements [2]. Based on these observations, it was hypothesized that the presence of an acidic component in the GIC liquid, such as tartaric acid, itaconic, or polyacrylic acid, led to a significant decrease in mineralization of the cells [16]. In a study by Bajantri et al. the cytotoxicity was higher in GIC followed by zinc phosphate, ZOE, and resin cement [17].

In the current study, dental pulp stem cells were derived from human dental pulp tissue, similar to other studies [18,19,20]. NIB cement positively affected cell proliferation in this study. This may be because the initial pH of the ZnP cement is nearly 2 or 3 owing to the presence of phosphoric acid and even after the final setting, it does not exceed a pH of 5 or 6. This creates a highly acidic environment that is unfavorable for the bone-forming cells [7]. The NIB cement contains mineral trioxide, calcium aluminate, and strontium fluoride, which have the least negative effect on the mineralization of osteoblasts [2]. Previous research reported the potential beneficial effects of two dental cements on the viability and proliferation of hDPCs. In addition, Zhang et al. showed that zinc-bioglass significantly promoted odontogenic differentiation and angiogenesis in hDPCs [18,19].

Cell proliferation was visible in both direct and indirect methods in this current study. Marvin et al. demonstrated low cell viability percentages upon direct contact with ZnP and NIB cements when cemented on commercially pure titanium [2]. Rodriguez et al. showed that zinc oxide non-eugenol dental cement affected the cellular host response of osteoblasts and gingival fibroblasts to a lesser extent than zinc oxide eugenol, ZnP, and acrylic resin cements [3].

The cytotoxic effects observed in this study highlight that both cement formulations; glass inomer cement and zinc phosphate cement induced moderate cytotoxicity in osteoblasts, as measured by ISO standards 10993-5 and 10993-12 [12]. The MTT and WST-1 assays provided reliable quantitative data on the metabolic activity and viability of the cells, confirming reduced viability in the presence of the cement formulations. Notably, primary osteoblasts were more sensitive to the formulations than MG-63 cells, which may be due to their closer resemblance to in vivo conditions. These results emphasize the importance of selecting biocompatible cement formulations to minimize adverse effects on bone-forming cells during dental restoration procedures.

The limitations of the study are, first, that the study was conducted in vitro. Second, the experiment did not simulate the exact intraoral environment of the human oral cavity. Finally, the mixing mechanisms of the three cements are different. The GIC and ZnP cements are available as powders and liquids and are hand-mixed, whereas the NIB cement is available as a capsule. Manual mixing allows the operator to adjust the powder–liquid ratio for the desired consistency, but with the pre-weighed powder and liquid in the capsule, the water–liquid ratio of the bio-ceramic cement cannot be modified by the clinician.

## 5. Conclusions

There is no definite criterion for the selection of dental cements for cement-retained implant restorations. This study determined the importance of the biocompatibility of dental luting cements and will help clinicians make informed choices during the selection of cements for the cementation of implant prostheses. The NIB cement showed the least negative effect on the mineralization of osteoblast cells compared to other routinely used test cements. However, further studies are needed to evaluate osteoblast and osteoclast activity in vivo.

## Figures and Tables

**Figure 1 medicina-60-01622-f001:**
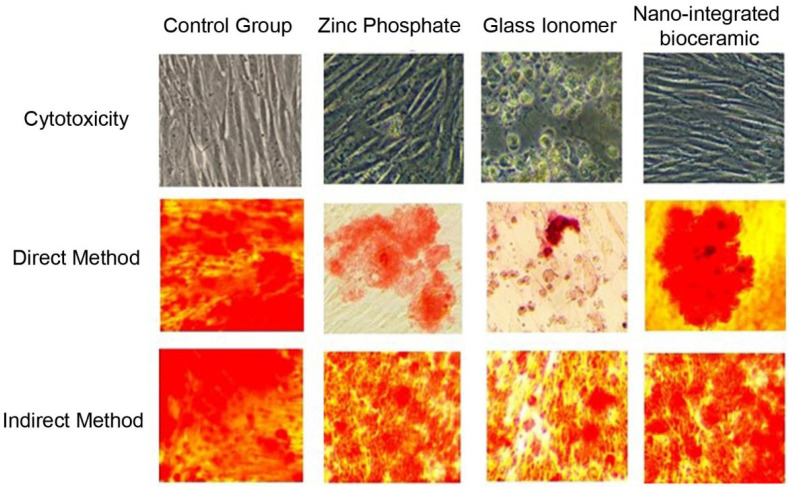
Cytotoxicity of cement formulations tested on primary human osteoblasts and MG-63 osteoblast-like cells using MTT and WST-1 assays.

**Figure 2 medicina-60-01622-f002:**
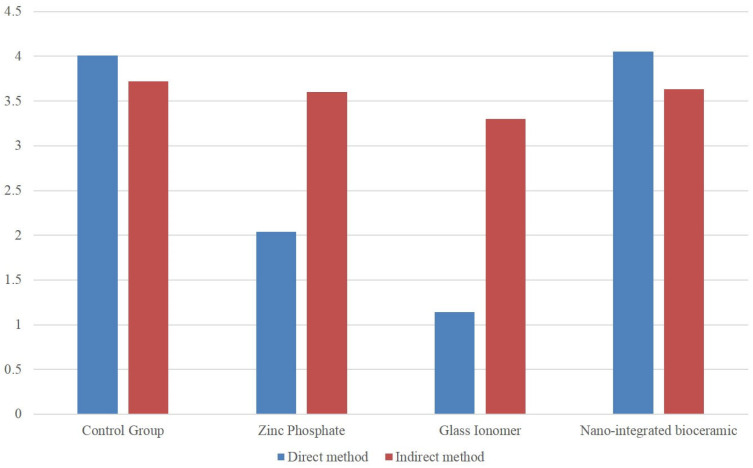
Comparison of different cements on hDPCs by direct and indirect method.

**Table 1 medicina-60-01622-t001:** Percentage of viable cells (mean ± SD) measured by MTT and WST-1 assays after exposure to different cement formulations. Both primary human osteoblasts and MG-63 cells were tested. Viability is expressed as a percentage relative to untreated control cells. Data represent the mean ± standard deviation (*n* = 3). Asterisks indicate significant differences from the control group (*p* < 0.05).

Cement Type	Group	*n*	Mean (Cell Viability %)	Std. Deviation	Mean Difference	t	*p*-Value
Glass ionomer cement	1 (MTT assay)	12	1.14	0.49	−2.15	−13.038	<0.001 *
	2 (WST-1 assay)	12	3.30	0.29			
Zinc phosphate cement	1 (MTT assay)	12	2.04	0.63	−1.71	−8.307	<0.001 *
	2 (WST-1 assay)	12	3.60	0.33			
Nano-integrated bio-ceramic cement	1 (MTT assay)	12	4.05	0.45	0.42	2.873	0.009
	2 (WST-1 assay)	12	3.63	0.22			
Control (no cement)	1 (MTT assay)	12	4.01	0.88	0.29	1.037	0.311
	2 (WST-1 assay)	12	3.72	0.41			

## Data Availability

The raw data supporting the conclusions of this article will be made available by the authors upon request.

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
