# Peer review of "Evaluation and Comparison of the Effect of Three Dental Luting Cements on Mineralized Bone Derived from Dental Pulp Stem Cells: An In Vitro Study"

_medicina, 2024, doi:10.3390/medicina60101622_

Round 1

Reviewer 1 Report

Comments and Suggestions for Authors

Dear Authors,

Respectfully,

Firstly, I should emphasize that this manuscript is relevant to the field and presented according to the demands of the journal.

When I read your manuscript for the first time, I was convinced that I saw a similar manuscript in Pubmed. However, nobody tested in vitro the effect of those three kinds of cement on mineralized bone derived from dental pulp stem cells. So, it could be considered as a novelty. The data are reproducible and well-understood.

The cements are very well explained and their preparation for the using in the experiment. For example, Dental cement also plays a crucial role in the retention of the abutment-crown interface in cement-retained implant prosthesis (CRIP). Clinicians select luting cements based on their physical, mechanical, and biological properties. I especially liked the sentence “The groups of cement tested in our study ranged from commonly used dental luting cement for permanent cementation of teeth and implant-supported prostheses (ZnP cement and GIC) to the one specifically designed for cement-retained implant prostheses (NIB cement)”.

However, the number of the techniques used is not quite clear. For example, you tested the effects of cement on the cytotoxicity of the cells. In the first figure, we can see that the authors tested cytotoxicity,  but you only identified the tests as International Organization for Standardization (ISO) methods 10993-5 and 10993-12. You could represent results quantitatively and clearly state if the cells are osteoblasts (as it says in the figure) or primary cells or you did it on both. The figures and table are not clearly explained.

The other important thing is using t-test for statistics when you have more than two groups. It was good to examine the effects of different types of cement on HDPCs by the direct and indirect method for individual samples on the mineralization of osteoblasts derived from dental pulp stem cells, but it would be interesting to inspect the effects between three different kinds of cement, and there you can't apply the t-test. On the other side, the results you achieved about the effects of cement on mineralization are expressed appropriately in the table and figures.

The limitations of the study, presented at the end of the discussion, are well-defined.

The conclusion summarises the results of the manuscript and the specific value of the conclusion is that the authors stated that their results can help clinicians to make informed choices of cements they are going to use. On the other hand, as I said some results should be added quantitatively, like the effects of cement on the cytotoxicity on DPSC or osteoblasts or both. And statistics could be more complex.

The references represent the text of the manuscript, however they could be updated.

Comments on the Quality of English Language

The English language is appropriate and understandable.

Author Response

Comments: Some results should be added quantitatively, like the effects of cement on the cytotoxicity on DPSC or osteoblasts or both. And statistics could be more complex.

Author's Response: Thank You for the feedback.

The results has been updated, and the statistics modified to explain the cytotoxicity.

All the updated references related to the research has been already incorporated.

Reviewer 2 Report

Comments and Suggestions for Authors

Table 1. does not describe how was the effect measured. In the text is also not clear. I consider the Table should be more descriptive

Figure 2. Almost the same as in Table 1. The scale shown in the graphic is not clear.

I would consider that the "effect" you mentioned in the text is not well described in the methodology section, which creates big confusion in the rest of the article and the figures and graphics. 

Comments on the Quality of English Language

In general, the level of english is easy to read and in my opinion no major corrections should be made. 

Author Response

Comment 1:  Table 1. does not describe how was the effect measured. In the text is also not clear. I consider the Table should be more descriptive.

Response 1: Thanks for the feedback. More descriptive modified Table 1 has been updated.

Comment 2:  Figure 2. Almost the same as in Table 1. The scale shown in the graphic is not clear.

Response 2: The necessary descriptive explanation has been provided.

Comment 3: I would consider that the "effect" you mentioned in the text is not well described in the methodology section, which creates big confusion in the rest of the article and the figures and graphics. 

Response 3: The necessary modifications has been updated in the methodology section and a detailed explanation has been added in the methodology section.

Reviewer 3 Report

Comments and Suggestions for Authors

Letter to authors,

I reviewed your interesting paper intitled “Evaluation and comparison of the effect of three dental luting cements on mineralized bone derived from dental pulp stem cells: An in vitro study” with interesting results. The materials and methods section with the methodology seems right but I believe that the Discussion section can be extended with more comments and literature data, and to better explain what value adds this paper to already published data, and to explain the strengths of the study (you only have limitations in the manuscript).

Author Response

Comments: Discussion section can be extended with more comments and literature data, and to better explain what value adds this paper to already published data, and to explain the strengths of the study.

Response: Thank you for your valuable feedback. The methodology section along with the strength of the study has been modified.